# The Alkylating Agent Methyl Methanesulfonate Triggers Lipid Alterations at the Inner Nuclear Membrane That Are Independent from Its DNA-Damaging Ability

**DOI:** 10.3390/ijms22147461

**Published:** 2021-07-12

**Authors:** Sara Ovejero, Caroline Soulet, María Moriel-Carretero

**Affiliations:** 1Institut de Génétique Humaine (IGH), Université de Montpellier-Centre National de la Recherche Scientifique, 34396 Montpellier, France; sara.ovejero-merino@igh.cnrs.fr; 2Department of Biological Hematology, CHU Montpellier, 34295 Montpellier, France; 3Centre de Recherche en Biologie Cellulaire de Montpellier (CRBM), Université de Montpellier-Centre National de la Recherche Scientifique, 34293 Montpellier, France; caroline.soulet@crbm.cnrs.fr

**Keywords:** methyl methanesulfonate, lipid stress, nuclear membrane, nuclear lipid droplets

## Abstract

In order to tackle the study of DNA repair pathways, the physical and chemical agents creating DNA damage, the genotoxins, are frequently employed. Despite their utility, their effects are rarely restricted to DNA, and therefore simultaneously harm other cell biomolecules. Methyl methanesulfonate (MMS) is an alkylating agent that acts on DNA by preferentially methylating guanine and adenine bases. It is broadly used both in basic genome stability research and as a model for mechanistic studies to understand how alkylating agents work, such as those used in chemotherapy. Nevertheless, MMS exerts additional actions, such as oxidation and acetylation of proteins. In this work, we introduce the important notion that MMS also triggers a lipid stress that stems from and affects the inner nuclear membrane. The inner nuclear membrane plays an essential role in virtually all genome stability maintenance pathways. Thus, we want to raise awareness that the relative contribution of lipid and genotoxic stresses when using MMS may be difficult to dissect and will matter in the conclusions drawn from those studies.

## 1. Introduction

The study of genome stability maintenance pathways frequently involves the use of tools to create DNA damage. These can be of a genetic nature, for example by removing a gene whose product protein is important for the repair of DNA lesions, which therefore accumulate. In addition, DNA damage can be created by external interventions such as the use of physical and chemical agents (genotoxins). Although the use of this latter strategy has granted an enormous power for assessing DNA-damage-related questions, it is not devoid of drawbacks, since these treatments are prone to harm other cellular molecules. For example, the widely used radiomimetic agent zeocin also affects components of the cytoskeleton [1], and hydroxyurea, regularly used to deplete the pool of dNTPs and trigger replication stress, is also recognized for its oxidative potential [2]. Apart from chemicals, physical approaches also trigger collateral effects. As such, creating single- and double-strand breaks in the DNA with ionizing radiation provokes the additional cleavage of sphingosine moieties at the plasma membrane, thus releasing ceramides, which can trigger apoptosis faster than the simultaneous breaks occurring in the DNA [3].

A frequently used agent in the field of genome integrity maintenance is the alkylating agent Methyl methanesulfonate (MMS). This molecule alkylates DNA bases, namely guanine to 7-methylguanine and adenine to 3-methlyladenine [4]. This way, the helix morphology and its information are distorted. Apart from the evident mutagenic potential, these lesions can provoke a myriad of secondary effects, many of which manifest during the process of DNA replication. For example, at modest doses, MMS can prevent the DNA polymerase from smoothly sliding over DNA, forcing its “jumping” forward, thus leaving unreplicated tracts behind, which equals single-stranded portions accumulating in the genome. At high doses, the block the alkylated moieties represent is so powerful that they literally halt the replication fork, a situation that, if not resolved, may lead to fork collapse and concomitant DNA breakage [5]. Given these abilities, MMS is a model agent to characterize the mechanisms with which the cell copes with alkylating lesions on its DNA, a field of immediate application in chemotherapy efficacy. Apart from this and depending on cell culture conditions, MMS lethality can become primarily ascribed to oxidation and not to DNA lesions [6]. Further, it can trigger a vast change in the cell proteome acetylation profile [7].

In this short report, we want to raise awareness of MMS ability to trigger a lipid stress at the inner nuclear membrane. We show that MMS elicits a growth of membrane structures that manifests differently depending on the basal rigidity of the nuclear membrane. This way, in plastic scenarios, the membrane expands and adopts a flower-like aspect. On the contrary, in expansion-resistant contexts, the stress is alleviated through the emission of nucleoplasmic reticulum protrusions and nuclear lipid droplets. Given the importance of the nuclear membrane in virtually all genome stability maintenance pathways, the consideration of these lipid transitions in the nuclear niche is of utmost relevance.

## 2. Results

We work to understand the crosstalk between genome integrity and lipid metabolism. During our studies to evaluate the potential impact of genotoxins on the formation of lipid droplets in RPE-1 (retinal pigmented epithelium) cells, we realized that 2 h treatment with the alkylating agent MMS induced the formation of nuclear lipid droplets (nLD) (Figure 1A,B). Lipid droplets are organelles shielded by a phospholipid monolayer that store lipids and arise from the endoplasmic reticulum (ER), in this case from the inner nuclear membrane (INM) subdomain and into the nucleoplasm [8]. The increase in nLD triggered by MMS was detected irrespective of the plotting strategy, either when considering the number of nLD-positive cells (Appendix A) or the number of nLD per positive cell (Appendix A).

We also exposed cells to other genotoxins, namely zeocin and hydroxyurea (HU), which trigger genotoxic stress in two different ways. Zeocin is a radiomimetic agent causing single- and double-strand breaks, whose impact can be monitored following the DNA damage response (DDR) effector Chk2 phosphorylation. HU leads to the exhaustion of the dNTP pool, thus halting the progression of replication forks, monitored by the progressive phosphorylation of the DDR effector Chk1. We conducted kinetic studies in response to these agents in comparison with MMS and established that, at 2 h, the extent of both phosphorylated Chk1 and Chk2 triggered by MMS was equal or inferior to that elicited by HU or zeocin, respectively (Figure 1C, red arrows). However, at this timepoint, the birth of nLD was not stimulated either by HU or zeocin (Figure 1B). This, along with the detection of comparable levels of DNA damage markers on chromatin, namely γH2AX and 53BP1, at 2 h treatment with zeocin or MMS (Appendix A), suggested that the ability of MMS to induce nLD formation probably was DNA-damage-independent.

We postulated that MMS may somehow trigger a lipid modification that culminates with the eviction of these lipids from its original location. This was reminiscent of the reported ability of various stresses, such as glucose deprivation, C2-ceramide addition, acidic pH, the ER stressor tunicamycin, or cerulenin to trigger the formation of LD [9]. Among them, we chose cerulenin, which directly relates to the lipid metabolism, given its ability to inhibit the fatty acids synthase, and measured for the first time its potential ability to induce nLD. Importantly, cerulenin treatment induced nLD formation modestly in RPE-1 cells (Figure 1B and Appendix A), yet its combination with MMS yielded an additive pattern of nLD that, again, was specific to MMS and not to other genotoxins (Figure 1B). Of note, independently of the number of nLD, this combination was remarkable when considering the profusion of the nLD entry into the nucleus, which even displaced the chromatin, as estimated by the lack of DAPI staining (Figure 1D, yellow arrowheads). Thus, by combining MMS and cerulenin, we could reveal a strong potential for nLD emission in RPE-1 cells.

RPE-1 cells display a nuclear shape that is consistently maintained irrespective of the treatment, as simply estimated by the constant form of the DAPI-defined nuclei (Figure 2A). Therefore, the emission of nLD could reflect a need for releasing a lipid stress from the nuclear membrane. In that case, the use of cells with a more flexible membrane would be expected to display a milder phenomenon. Cancer cells present a more deformable membrane, for example capable of constriction to go through very narrow channels [10]. Another feature illustrating their flexibility is the easiness with which the nuclei of these cells form blebs, which further give rise to micronuclei during interphase [11]. We chose HepG2 cells, derived from hepatocarcinoma, to further support the notion that nLD birth in response to MMS relates to the rigidity of the nuclear membrane in RPE-1 cells. Our working hypothesis was that, provided that these cells deform their nuclear perimeter in response to MMS, this would be paralleled by a milder or even absent emission of nLD. First, MMS triggered a neat deformation of HepG2 nuclei shape (Figure 2A). However, contrary to the prediction, in response to MMS (Figure 2B), HepG2 cells significantly emitted nLD (Figure 2C). Importantly, while cerulenin alone also triggered nLD birth, as in RPE-1 cells, this emission was not further increased when combining it with MMS (Figure 2C and Appendix A). Thus, the ability of HepG2 to increase nLD generation is more limited in a system where membrane deformation is possible.

Of note, the emission of micronuclei, which operates toward the cytoplasm, was undetectable in RPE-1 cells and irresponsive to MMS and/or to cerulenin in HepG2 (Figure 2D). This suggests that the ability of MMS to trigger the deformation of nuclear shapes (in HepG2 cells) or to give rise to nLD (in RPE-1 cells) was probably associated to the INM. It is therefore likely that lipid stress generated in the INM by MMS is alleviated by a combination of nuclear membrane deformation and emission of nLD, depending on the nuclear membrane basal flexibility.

Lastly, we reasoned that, if MMS elicits lipid transitions, which in HepG2 can deform the membrane, while in RPE-1, cells are forced to transit toward the nucleoplasm, additional membrane-related events were likely to be detected. We considered the growth of the nucleoplasmic reticulum (NR), an organelle derived from the nuclear membrane that invaginates toward the nucleoplasm [12]. We used immunofluorescence to detect the NR marker SUN1 [13] followed by image convolution to monitor the eventual growth of this organelle upon MMS treatment in RPE-1 cells and, as a control, in HepG2. Importantly, we indeed detected these events exclusively in RPE-1 cells (Figure 3A) and in response to MMS treatment (Figure 3B, compare with + zeocin). These structures appeared to migrate from the nuclear periphery toward the nucleoli (Figure 3A, MMS, yellow arrows), a “destination” previously reported for these bodies [12,14].

Pretreatment of cells with cerulenin did not abrogate these MMS-elicited structures, but instead disoriented their migration, rendering them parallel to the nuclear periphery (Figure 3A, cerulenin > MMS, yellow arrows; and Figure 3B). As further support to the notion that MMS-elicited lipid structures mostly stem from the INM, we hardly detected (only once or twice per experiment) NR-reminiscent structures positive for the outer nuclear membrane marker calnexin [15] (Appendix A, MMS, red arrows).

Overall, we report that the alkylating agent MMS has the ability to trigger lipid transitions, most probably unrelatedly to its DNA-damaging ability. These modifications cause a membrane stress, likely at the INM, that needs to be released. In cells with potential for nuclear membrane flexibility (such as HepG2), this results in nuclei adopting a flower-like aspect and a limited number of nLD. However, in rigid nuclear membrane contexts (such as in RPE-1 cells), this stress is alleviated through the emission of both nLD and NR arms (Figure 4).

## 3. Discussion

DNA-damaging agents are chemical entities with multiple cellular targets. The agent used in this work, MMS, is an alkylating molecule that acts on DNA by preferentially methylating guanine and adenine bases. MMS is broadly used in the research field of genome stability, both for fundamental studies and as a model for mechanistic studies to understand how alkylating agents used as chemotherapeutics work. Its additional effects concern its strong oxidizing potential which, depending on the cell culture conditions, may be the leading reason of death rather than its harm to the genetic material [6]. Additionally, it exerts a broad change in protein acetylation [7]. In this work, we introduce the notion, not to be neglected, that it can also trigger a lipid stress that stems from and affects the inner nuclear membrane. Given the intricate relationship between this location and genome stability (see next paragraph), we warn that the relative contribution of lipid and genotoxic stresses when using MMS may be difficult to dissect.

The nuclear membrane not only encloses the genome, thus separating it from the rest of the cell, but also acts as a scaffolding anchor susceptible of impacting every nuclear transaction. For example, the nuclear membrane ensures the coordination of transcription and replication, during which the DNA needs to be dynamically anchored to and dislodged from the nuclear membrane [16]. Anchor of telomeres to the nuclear membrane instructs their silencing [17] and warrants their length maintenance [18]. In the same line, the nuclear membrane influences the establishment of chromosome territories, of the epigenome and its maintenance [19,20,21], it controls the DNA-damage response [22,23] and DNA repair [13,24,25]. In a more incipient field, the lipid composition of the nuclear membrane directly conditions cell fitness in the case of aneuploidy [26], and overgrown membranes can trap thus prime the mis-segregation of chromosomes [27]. Excitingly, the naturally polyploid cells from *C. elegans* intestines also form nLD and NR arms derived from the INM that travel toward nucleoli [28]. These arms, as the ones we show (Figure 3A), could have additional impacts on nuclear homeostasis. First, these structures have already been implicated in assisting the repair of double-strand breaks happening in the ribosomal DNA [13]. Second, ER membranes have very recently been shown to embrace, ivy-wise, membrane-less condensates such as P-bodies and stress granules, splitting them in two [29]. Given that nucleoli are the biggest nuclear membrane-less condensates, it is tempting to imagine that NR membranes could execute the fission of nucleoli. All in all, the lipid alterations fostered by MMS, whether restricted to the INM or extended toward the nucleoplasm, are very likely to impact genome stability by other means than through the direct damage to the DNA.

The differential resistance to membrane expansion observed between RPE-1 and HepG2 cells could stem from their nontransformed versus cancerous origins, respectively. Indeed, the nuclear membrane of cancer cells is acceptedly described as prone to deformation and of higher plasticity [10]. Along this line, a feature such as their easiness for nuclear blebbing and micronuclei formation toward the cytoplasm during interphase is well documented [11]. However, the phenomenon we describe here is more likely to relate to events occurring at the INM. In this sense, an alternative explanation could relate to the cell-type-specific pathways ruling phosphatidylcholine (PC) synthesis. HepG2 cells, of liver origin, have two different active pathways for the synthesis of this central phospholipid. As all human cells, they possess an active Kennedy pathway, through which CDP-choline esterified with DiAcylGlycerol gives rise to PC. However, a 30% of their PC synthesis is ensured by the PEMT pathway, in which PC is produced from phosphatidic acid (PA) in a process involving multiple methylations [30]. Perhaps, this ability endows them with a wider window for generating PC, thus more membrane expansive properties, capable of buffering whatever the stress imposed by MMS. By contrast, RPE-1 cells, lacking channeling through the PEMT pathway, may accumulate an upstream intermediate giving rise to NR overgrowth and nLD formation. This is a tempting hypothesis because, as mentioned, such an intermediate could be PA, whose ability to support negative curvature at membranes would be perfectly positioned to rule these transitions. This hypothesis is further supported by the fact that PA is closely related to the formation of nLD [31,32,33] Together, we bring the notion that the crosstalk between lipids and genome instability not only concerns physiological processes but may be also extended to the tools with which we tackle their study.

## 4. Materials and Methods

### 4.1. Reagents

NileRed (HY-D0718, CliniSciences, Nanterre, France), DAPI (D9542, Sigma-Aldrich, Saint-Quintin Fallavier, France), Hoechst (B2261, Sigma-Aldrich, Saint-Quintin Fallavier, France), cerulenin (SC-200827A, Santa Cruz Biotechnology, Dallas, TX, USA), Methyl methanesulfonate (29925, Sigma-Aldrich, Saint-Quintin Fallavier, France), Zeocin (R25001, ThermoFisher, Illkirch-Graffenstaden, France), Hydroxyurea (H8627, Sigma-Aldrich, Saint-Quintin Fallavier, France), ProLong (P36930, ThermoFisher, Illkirch-Graffenstaden, France).

### 4.2. Antibodies

Antibodies used in this work were anti-SUN1 (ab124770, Abcam, used at 1/500); anti-nucleolin (ab136649, Abcam, used at 1/200); anti-Chk1 (sc8408, SantaCruz Biotechnologies, used at 1/500); anti-P-Ser345-Chk1 (2348LS, Cell Signaling, used at 1/1000); anti-P-Thr68-Chk2 (2661S, Cell Signaling, used at 1/1000); anti-Chk2 (05-649, Sigma-Aldrich, used at 1/1000); anti-calnexin (610523, Becton-Dickinson, used at 1/1000 for Western Blot and at 1/100 for immunofluorescence); anti-γ(pSer139]H2AX (05-636, Sigma-Aldrich, used at 1/500); anti-53BP1 (NB100-304, Novus, used at 1/500).

### 4.3. Human Cell Culture and Treatments

RPE-1 and HepG2 cells were a kind gift from Krzysztof Rogowski and Urszula Hibner, respectively. They were seeded on p100 plates (dilution 1:7 from one confluent p100). After two days, medium was changed (at 37 °C), and treatments were applied. RPE-1 were grown in DMEM (D5796-500ML, Sigma, Saint-Quintin Fallavier, France) and HepG2 in DMEM + GlutaMAX (Gibco, 31966-021, ThermoFisher, Illkirch-Graffenstaden, France) supplemented with 10% FBS (S1810-500, Biowest, Nuaillé, France) and 1% Pen/Strep (P0781, Sigma, Saint-Quintin Fallavier, France). At the end of the corresponding treatments, cells were collected for Western blot or fixed for immunofluorescence with 4% PFA/PBS (20 min at room temperature), and then washed once with PBS. For SUN1, a pre-extraction was done putting plates on ice with 20 mM NaCl, 0.5% NP-40, 20 mM Hepes pH 7.5, 5 mM MgCl_2_, 1% DTT for 20 min prior to fixation. We verified that this did not affect nucleolin patterns. DNA-damaging treatments were always applied for 2 h at a final concentration of 0.005% MMS, 100 μg/mL zeocin or 1 mM hydroxyurea. Pretreatment with cerulenin was done for 4 h with 5 μg/mL cerulenin (2 h prior to addition, or not, of the genotoxic agents).

### 4.4. Immunofluorescence

Fixed cells on coverslips were washed once with PBS and permeabilized with 0.2% Triton/PBS for 10 min at room temperature, then saturated with 3% BSA/PBS for 30 min. Coverslips were incubated with primary antibodies diluted in 3% BSA/PBS for 90 min then washed 3 times x 10 min with PBS under gentle shaking. Coverslips were further incubated with fluorophore-coupled secondary antibodies diluted (1:500) in 3% BSA/PBS for 45 min while protecting them from light from this point on, then washed 3 × 10 min with PBS under gentle shaking, then incubated for 5 min with Hoechst (20 μg/mL) or for 10 min with DAPI (1 μg/mL), both diluted in H_2_O, and washed 3 times with H_2_O. Finally, coverslips were allowed to dry and mounted using ProLong, then left to dry overnight at room temperature in the darkness.

### 4.5. Western Blot

All the samples were lysed with high salt buffer (50 mM Tris pH 7.5, 300 mM NaCl, 1% Triton X-100, protease inhibitors (Roche), phosphatase inhibitors Halt cocktail) (400 μL per p100) for 10 min on ice with frequent vortexing. After lysis, samples were centrifuged 10 min at 14,000 rpm at 4 °C, and supernatants retrieved for Western blot. Typically, 20–30 μg of whole cell extracts was loaded onto homemade 10% acrylamide gels (1.5 mm thick) and migrated 70 min at 40 mA per gel (migration buffer: 25 mM Tris, 200 mM glycine, 0.1% SDS). The proteins were transferred to a nitrocellulose membrane for 2 h at 100 V in transfer buffer (25 mM Tris, 200 mM glycine, 20% ethanol).

### 4.6. Image Analyses, Quantification and Statistics

Fluorescent signals were detected using adequate wavelengths and acquired either with a Zeiss CCD AxioCam MRm monochrome camera from a Zeiss AxioImager Z1 microscope with ApoTome technology or with a Zeiss Axioimager Z2 microscope, in both cases using a 40× Plan Apochromat 1.3-NA oil objective lens and either Zen or Metamorph software, respectively. Images were acquired at 20–23 °C. Subsequent image visualization, co-localization, and analysis were performed with Image J v2.0.0-rc-69/1.52i. Both the number of cells with micronuclei and the number of nLD per cell were counted visually by the experimenter. The processing of SUN1 signals was conducted by choosing the “Process > Filter > Convolve” tool without changing the basal parameters. Graphical representations and statistical analyses were conducted by using GraphPad Prism v9. Given that the studied populations did not appear as normally distributed, the significance of possible differences between the populations is indicated by asterisks and *p*-values obtained after applying a Mann–Whitney nonparametric test in all instances.

## Figures and Tables

**Figure 1 ijms-22-07461-f001:**
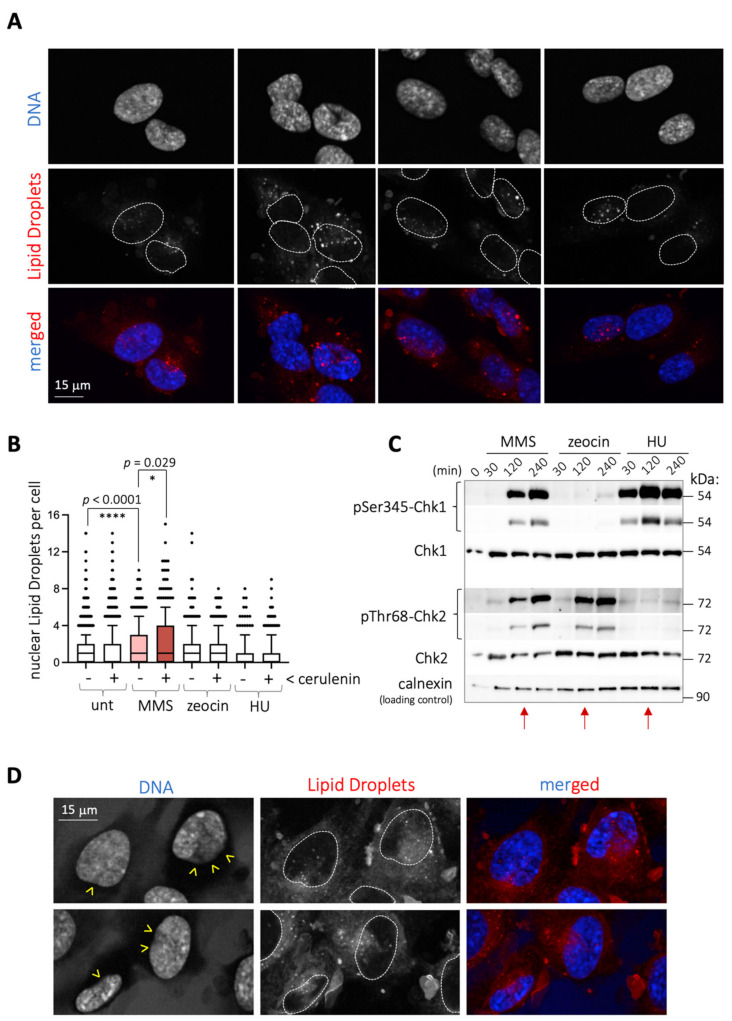
MMS triggers the formation of nuclear LD in RPE-1 cells. (**A**) Microscopy images corresponding to RPE-1 cells treated for 2 h with 0.005% MMS. Nuclei are identified by staining DNA with DAPI, and LD with the vital dye NileRed. To facilitate visualization of the respective position of the LD within the nucleus, merged images are shown at the bottom, and the perimeter of nuclei superimposed as dashed lines onto the LD images. (**B**) Quantification of the number of nuclear LD counted in RPE-1 cells upon treatment with the indicated agents. When cerulenin is indicated, it was added at 5 μg/mL to the cultures for 4 h, plus the incubation or not with 0.005% MMS, 100 μg/mL zeocin or 1 mM HU during the last 2 h. Each dot corresponds to the value noted for one nucleus. At least 200 nuclei per experiment and condition were counted from 3 independent experiments. The boxes comprise the 10–90 percentile values, and their central bar represents the median of the population. Since populations were not normally distributed, the significance of differences between the populations is indicated by asterisks (*) and their associated *p*-values, obtained after applying a Mann–Whitney nonparametric test. (**C**) Western blot of extracts from RPE-1 cells untreated or treated with either 0.005% MMS, 100 μg/mL zeocin or 1 mM HU for the indicated times to monitor the activation of the DDR by following the phosphorylation on Serine 345 (pSer345) of the effector Chk1 and that of Chk2 on threonine 68 (pThr68). Molecular weights are shown in kDa as indicated by co-run molecular weight ladders. Red arrows mark the samples chosen for the microscopy studies shown in (**B**). (**D**) Details as in (**A**) for cells treated 4 h with cerulenin at 5 μg/mL, and then 0.005% MMS added during the last 2 h. Dashed white lines define the nucleus boundaries. Yellow arrowheads point at the locations of the nuclei periphery from which LD are seen to access the nucleus, eventually preventing the DAPI staining.

**Figure 2 ijms-22-07461-f002:**
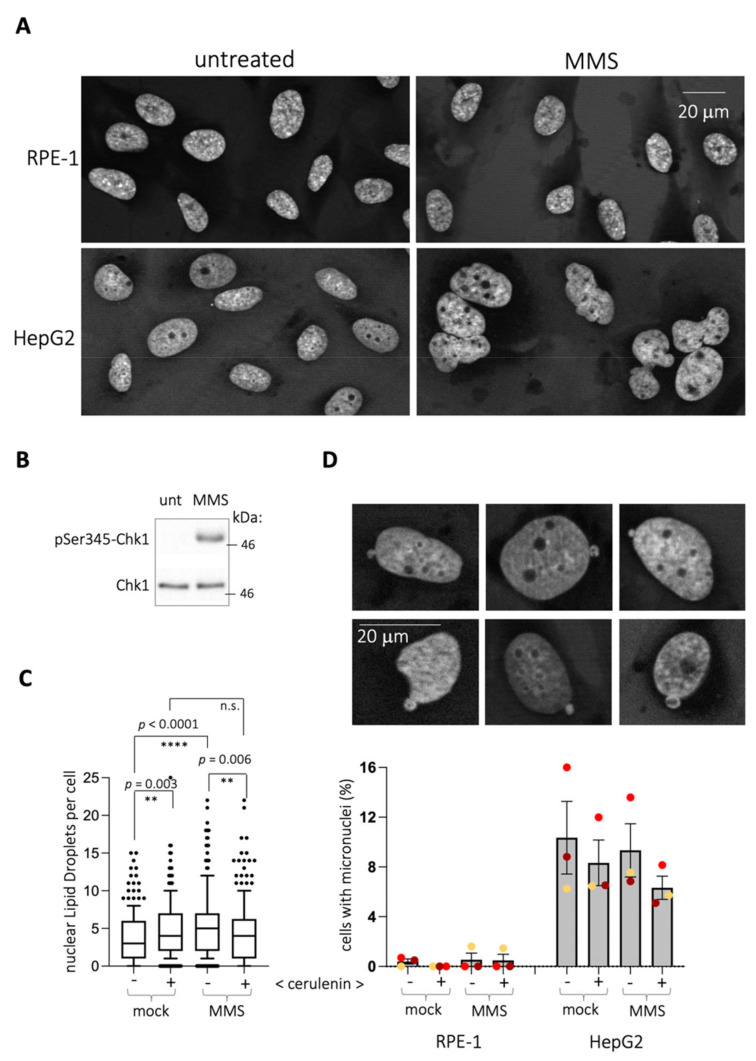
MMS triggers nuclear membrane alterations in HepG2 cells. (**A**) DAPI-stained nuclei of the indicated cell lines untreated or treated with 0.005% MMS for 2 h are shown to illustrate that MMS treatment in HepG2 triggers a general deformation of the nuclear perimeter. (**B**) Western blot of extracts from HepG2 cells untreated or treated with 0.005% MMS for 2 h to monitor the activation of the DDR by following the phosphorylation on serine 345 (pSer345) of the effector Chk1. Co-run molecular weight is indicated in kDa. (**C**) Quantification of the number of nLD counted in HepG2 cells upon treatment with the indicated agents. When cerulenin is indicated, it was added at 5 μg/mL to the cultures for 4 h, plus 0.005% MMS or not for the last 2 h. Each dot corresponds to the value noted for one nucleus. At least 200 nuclei per experiment and condition were counted from 3 independent experiments. The boxes comprise the 10–90 percentile values, and their central bar represents the median of the population. Since the populations were not normally distributed, the significance of differences between the populations is indicated by by asterisks (*) and their associated *p*-values, obtained after applying a Mann–Whitney nonparametric test. n.s. stands for “not significant”. (**D**) **Upper panel:** illustrative pictures of HepG2 DAPI-stained nuclei displaying blebbing of the DNA material toward the cytoplasm, the initiating events of micronuclei during interphase. **Lower panel:** quantification of the percentage of cells in the population displaying blebbing and/or close-to-the-nucleus micronuclei in the indicated cell types after the indicated treatments. Each colored dot corresponds to an independent experiment. At least 150 cells were counted per experiment and condition. Within a same cell line, none of the treatments led to any significant difference with respect to the other ones after applying a Mann–Whitney nonparametric test.

**Figure 3 ijms-22-07461-f003:**
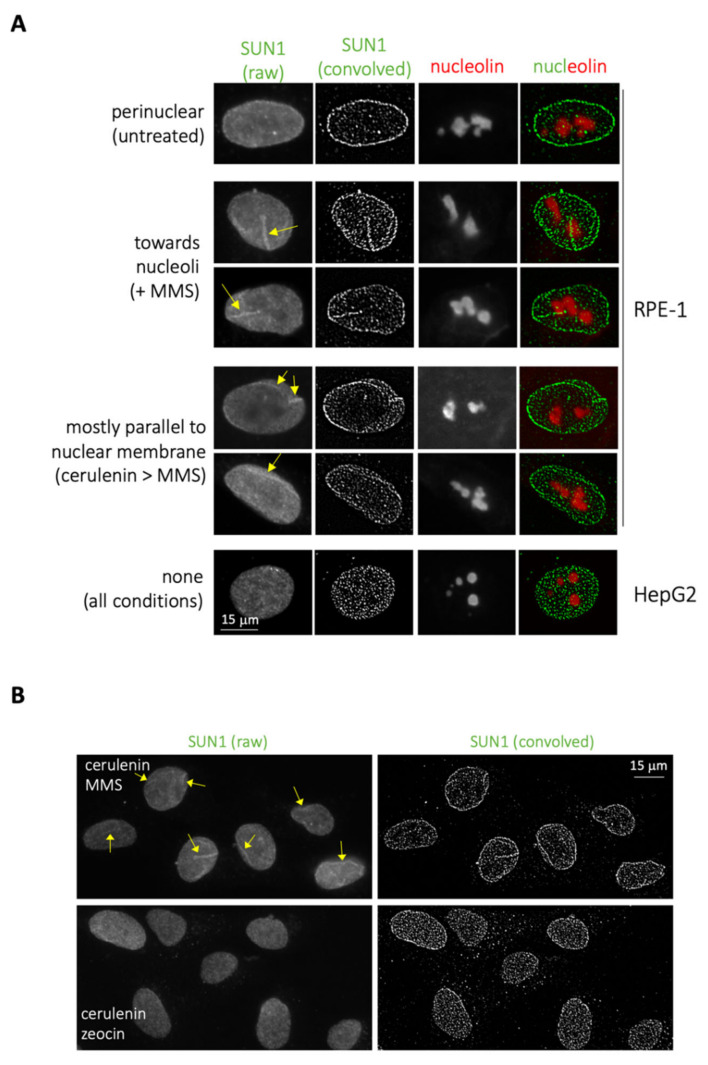
Increase in intranuclear membrane pools occurs in RPE-1 cells in response to MMS. (**A**) Immunofluorescence performed on RPE-1 and HepG2 cells to detect SUN1 and nucleolin signals. Visualization of membrane-associated SUN1 signals is enhanced by convolving the raw image. In RPE-1 cells, in untreated conditions, SUN1 signals are mostly perinuclear; in response to MMS, SUN1 invaginations reach nucleoli; preincubating the cells with cerulenin prior to MMS addition, SUN1 invaginations mainly run parallel to the nuclear membrane. In HepG2 cells, SUN1 antibody provided no membrane-associated pattern under these immunofluorescence conditions. (**B**) Immunofluorescence to detect SUN1 performed on RPE-1 cells treated 4 h with 5 μg/mL, the last 2 h either with additional 0.005% MMS or 100 μg/mL zeocin. On the left, raw images are shown, and NR signals are indicated by yellow arrows. On the right, the same images were subjected to the ImageJ tool “convolve”.

**Figure 4 ijms-22-07461-f004:**
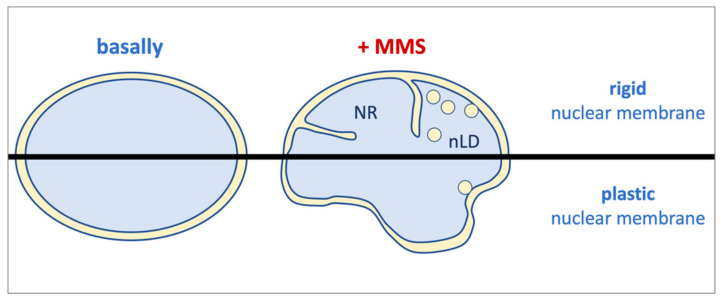
Model. Under basal conditions, nuclei display their ellipsoidal shape as ruled by their shielding membrane. Upon MMS treatment, a lipid stress is elicited at the inner nuclear membrane whose outcome is determined by the plasticity of such membrane. This way, in rigid contexts, such as in RPE-1 cells, the resistance of the membrane is alleviated by the emission of bodies toward the nucleoplasm, such as nuclear lipid droplets (nLD) and nucleoplasmic reticulum arms (NR). By contrast, in malleable scenarios such as in HepG2 cells, the membrane mostly responds by suffering an expansion, probably due to increased phospholipid synthesis, which makes it adopt a flower-like shape. Eventually, it can also emit nLD, albeit more modestly.

## Data Availability

All the data concerning this work are included within this manuscript or attached as Appendix A.

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
