# Peer review of "The Alkylating Agent Methyl Methanesulfonate Triggers Lipid Alterations at the Inner Nuclear Membrane That Are Independent from Its DNA-Damaging Ability"

_ijms, 2021, doi:10.3390/ijms22147461_

Round 1

Reviewer 1 Report

In this manuscript, the authors present data indicating that treatment of cells with the well-studied alklyating agent and mutagen MMS induces changes in lipid deposits at the nuclear membrane. Changes in lipid and membrane biochemistry are often overlooked in studies of genotoxicity, so the authors' work is potentially important. However, I have a number of concerns that detract from the ability of the authors to make convincing conclusions.

1) The authors state that their observations of nLD formation are independent of DNA damage because they do not observe effects with HU/zeocin at a single concentration and time point. Could the kinetics of the response be different for for these other agents? The authors use Chk1 phosphorylation as a marker of DNA damage, though I would argue that this is more of a marker of replication stress and does not necessarily indicate DNA damage in the absence of additional assays. Thus, comet assays or some other assay could be used to validate that their treatments are indeed causing DNA damage. Alternatively, showing that loss of DNA repair does not impact nLD formation could also provide evidence that the authors' observations are indeed independent of DNA damage. In the absence of doing additional experiments, one solution would be to remove the "that are Independent from its DNA-Damaging Ability" from the title.

2) In the text (line 181), the authors state that the lipid modifications "likely occur at the inner membrane". But in the title, the authors state "at the nuclear membrane". Are there additional experiments that can be done to convince the reader that this more than just "likely" at the INM?

3) The authors quantify the number of nLD per cell in their assays. From the data presented in Figure 1A, it looks like some cells have many (>10?) nLDs, but the quantification in Figure 1B indicates that average number of nLDs per cell is very low (about 1). I wonder if if it would be helpful to the reader if they were also provided with the percentage of cells that show >1 nLD under these conditions. Would this quantitation show a more dramatic difference between groups?

4) On line 96, "dying DNA with DAPI" sounds awkward. Use "staining"? 

5) It may be helpful to the reader if the magnification used in their microscopy images was provided in the methods section (along with details about microscope used?)

Reviewer 2 Report

This is a largely very well written manuscript, detailly describing applied methods, obtained results that are clearly presented although figure legends could be more concise. The only question to answer is whether this manuscript is scientifically sound. Pleiotropy of the action of any compound in any context is not a very surprising effect. I think that nobody claims that the action a DNA-damaging agent is limited to DNA damage, as this damage itself may initiate of a domino-like cellular reaction. In this context, the manuscript of Ovejero et al. does not show a landmark phenomenon. However, it convincingly presents formation of lipid droplets by MMS in two cell lines.

The authors describe some effects induced by MMS, but I do not understand why they insist on underlying pleiotropy in its action. It is not clear for me. They further write that other genotoxic agents do not induce lipid droplets. And so? MMS is the only one? Is it exceptional? Why? I think that the authors should focus on the effects they obtained, and the pleiotropy aspects should be just mentioned (2-3 sentence) in Discussion.

The title is not very clear. What do the author understand “lipid alterations independent of DNA-damaging ability”? The authors are right that MMS is often used to induce DNA damage response, but I think that this is not its only action. Furthermore, could be lipid alterations dependent on DNA damage? Directly? How?

What does it mean “potential link between genome integrity and lipid metabolism”? Why potential? Genome integrity is important for all cellular processes. I suggest rewriting this sentence and change the general attitude. I think that the first paragraph of the discussion section is redundant as it contains several obvious statements. On the other hand, the last sentence in Discussion is not clear for me – what is “the intimate link between lipids and genome instability”? I think that the authors did not shown even a solid association between the production of lipid droplets and genome instability as presented in Fig. 1 Chk1 phosphorylation alone does not determine genomic instability – some other markers should be included.

Why did the authors use RPE-1 cells? Retinal pigment epithelium cells in vivo are quiescent due to steric constrains in the retina. They start to proliferate in culture, but their relatively high fraction may display senescence as it was shown in many studies, excluding these cells from some studies to determine some physiological parameters. I understand that it was due to nuclear membrane shape in these cells that were compared with HepG2 cells.

It is not necessary to provide exact values of p in figures – stars would be enough, p < 0.0001 (four stars) is rather unusual – I suggest limiting to p < 0.001 (three stars)

Round 2

Reviewer 1 Report

The authors have largely address my concerns.

Reviewer 2 Report

Thank you for your explanation. I still suggest to limit to one-three stars in graphs - do not be a slave of the Prism software.